# Effects of Different Straw Mulch Rates on the Runoff and Sediment Yield of Young Citrus Orchards with Lime Soil and Red Soil under Simulated Rainfall Conditions in Southwest China

**Zechao Gao** [1,2,3], **Qinxue Xu** [2,4,*], **Qin Si** [1,2], **Shuaipu Zhang** [1,2], **Zhiyong Fu** [4] **and Hongsong Chen** [4]

1 Key Laboratory for Theory and Technology of Environmental Pollution Control in Guangxi, Science and Technology Innovation Base, Guilin University of Technology, Guilin 541004, China; zechaogao@yeah.net (Z.G.); siqin47@yeah.net (Q.S.); shuaipuzhang@glut.edu.cn (S.Z.)
2 Guangxi Collaborative Innovation Center for Water Pollution Control and Safety in Karst Area, Guilin University of Technology, Guilin 541004, China
3 Shaanxi Provincial Land Engineering Construction Group, Xi'an 710075, China
4 Huanjiang Observation and Research Station for Karst Ecosystems, Chinese Academy of Sciences, Huanjiang 547100, China; zyfu@isa.ac.cn (Z.F.); hbchs@isa.ac.cn (H.C.)
* Correspondence: xqx@glut.edu.cn

**Abstract:** Soil erosion has been a major environmental issue in young citrus orchards in karst areas of Southwest China. Straw mulching is an effective measure to reduce soil erosion. However, few studies have considered this measure in soil and water conservation in citrus orchards in karst areas. In this study, the effects of straw mulching on runoff and sediment reduction in young citrus orchards were studied using the simulated rainfall method with two typical soils—red soil and lime soil—in karst areas. This study consisted of two rainfall intensities (60 and 120 mm/h) and four straw coverages (0, 20, 50, and 80%). The results showed that straw mulching can significantly reduce surface soil loss in both lime soil and red soil. The surface runoff reduction rate of lime soil with straw mulching was much higher than that of red soil. Under the condition of heavy rainfall (120 mm/h), the surface soil loss rate was reduced by more than 57.5% in lime soil with 20% straw mulching. In red soil, a similar reduction rate of soil loss can only be achieved when straw mulching is 50%. Straw mulching reduced the surface soil loss of lime soil in young citrus orchards, mainly because of limited runoff transport, while red soil was limited by soil stripping. The results can provide a scientific basis for soil and water conservation in young citrus orchards in karst areas.

**Keywords:** simulated rainfall; soil erosion; orchard; conservation tillage; karst area

## 1. Introduction

China has the largest karst area (3.443 million km$^2$) in the world. Among the three concentrated-distribution karst areas in the world, Southwest China has the largest area of contiguous exposed carbonate rocks and the strongest karst development [1]. Northwest Guangxi, located in the transition zone from the Yunnan–Guizhou Plateau to the Guangxi Basin, is a typical karst area with a fragile ecological environment [2]. Due to the large population but less-cultivated land, many slopes in karst areas have been reclaimed for planting corn, rice, and other crops. With the development of the social economy and the management of the ecological environment, some of the abandoned reclaimed sloped lands have been restored to natural shrub grassland, but many sloped lands are used for planting fruit trees with high economic value such as forests or citrus.

Citrus orchards are common in karst areas of Southwest China. However, at the beginning of citrus planting, the surface of the orchard is bare, and serious soil erosion has become one of the main obstacles to agriculture development. Especially during rainstorms,

the generation of surface runoff causes many discrete effects, such as soil and water loss and nutrients being transferred to surface water bodies [3]. Tu A.G. conducted long-term observations of soil erosion in citrus orchards, and the results showed that the soil erosion intensity in citrus orchards can reach extremely high levels during the young tree stage [4]. The runoff coefficient was above 0.3 and soil erosion was serious at the beginning of citrus planting in karst areas of Spain [5].

Straw mulching is a typical practice in conservation tillage. Much research has been conducted on the effects of straw mulching on soil characteristics (the stability of soil aggregates, soil structure, and surface closure), hydrological responses (changes in soil moisture, infiltration, and runoff), soil loss, and sediment concentration [6–13]. Recent studies have shown that straw mulching can reduce soil splash erosion by absorbing kinetic energy during rainfall [14]. Several researchers have pointed out that straw mulching also maintains soil moisture on the surface, facilitates organic matter production, and reduces total surface runoff [15–18]. The changes in runoff and sediment yields largely depend on the antecedent soil conditions and topographic characteristics [19–25]. Straw mulching can reduce soil loss on the plot scale [26]. DeHaan simulated the effect of different mulch applications on reducing soil erosion and found that the runoff yield with straw mulching was 13 times lower than that of bare soil [27]. Poulenard et al. used a rainfall simulation to explore the impact of mulch coverage on soil erosion in cultivated land restoration after a fire, and they indicated that sediment loss was very low when the cover was used in the first year after a fire [28]. Adekalu pointed out that wolf grass cover could significantly reduce soil erosion, which increased with slope gradients [8]. Groen showed that using straw mulch with a coverage of 2.24 t·ha$^{-1}$ reduced soil loss after wildfires in northwest Montana [29]. Jiang reported that wheat straw cover reduced soil erosion by 95% compared with bare soil [30].

Zhang classified the peak-cluster depression area into non-karst areas dominated by clastic rocks and karst areas dominated by carbonate rocks [31]. The soil erosion process in karst areas is obviously different from that in non-karst areas: the surface runoff and the amount of surface soil loss are low, and there are certain characteristics of underground leakage [32]. Due to the difference in parent rock properties, soils in the same area are developed into many different types, which are also similar to the Houzhaihe small watershed of Puding County, Guizhou Province [33]. Eppes said, in the study of the Transverse Ranges of California, USA, that the differences in soil development will cause significant differences in terrain performance [34]. Lime soil and red soil are the main soil types in Northwest Guangxi, China. Red soil is formed by the weathering of carbonates or rocks containing other iron-rich aluminum oxides in hot and humid climates. Red soil usually has a deep red soil layer, with the obvious development of a reticulate layer. Clay minerals are mainly kaolinite, acidic, and have low base saturation. Red soil is not easily damaged by rain erosion, so red soil develops well under the leaching of rain. Lime soil is the soil developed by the limestone parent material. The texture of lime soil is generally sticky, and there are lime foam reactions in the section. It often has red, yellow, brown, black, and other colors. China's lime soil is mainly distributed in the southern subtropical region. There are significant differences in soil thickness and their physical and chemical properties. The benefits of straw mulching in different soil types for soil and water conservation are significantly different. Rahma's research showed that straw mulch can induce greater soil losses from loess slopes than no mulch under extreme rainfall conditions [35]. Somchai's research on sandy loam soil in northern Thailand showed that straw mulch can effectively reduce surface runoff and surface soil loss. Among them, a straw mulch of 5.0 t/ha has the strongest soil and water conservation benefits [36]. The main reason for the difference in the soil and water conservation effect caused by straw mulching in different soil types is that different soil types have different thicknesses, physical and chemical properties, bulk densities, and soil infiltration rates. Soil with a low soil infiltration rate and a high content of silt and sand is more prone to water and soil loss. There is still a lack of research making

a direct comparison of straw mulching's effects on the flow and sediment reduction in different soil types.

At present, citrus orchards in karst areas of Southwest China are rapidly expanding, but there is little research into the soil erosion process at the beginning of citrus orchards. We urgently need to study whether straw mulching has the same effect on water and soil conservation in young citrus orchards under different soil environments in the same region. To determine the impact that rice straw mulching has on the runoff and sediment reduction in a young citrus orchard with different soil types, four straw coverages (0, 20, 50, and 80%) and two rain intensities (60 and 120 mm/h) were set in the simulated rainfall experiment. The study was performed to verify two main hypotheses: (1) straw mulching could reduce surface runoff and soil erosion and (2) the effect of straw mulching on reducing surface erosion in lime soil and red soil could differ. Relevant conclusions will deepen the understanding of soil erosion mechanisms on sloped land in karst areas and provide a reference for water and soil conservation measures.

## 2. Materials and Methods

### 2.1. Study Area

The study area was the Mulian Comprehensive Experimental Demonstration Park of the Huanjiang Karst Ecosystem Observation and Research Station of the Chinese Academy of Sciences (Figure 1). The demonstration park is located in Dacai Township, Huanjiang Maonan Autonomous County in the northwest of the Guangxi Zhuang Autonomous Region, China (latitude 24°43′59″–24°44′49″, longitude 108°18′57″–108°19′58″). This area is located in the slope zone between the Yunnan–Guizhou Plateau in northwest Guangxi and the karst plain in central Guangxi, and the surrounding mountain stretch. The altitude is 272.0 m~647.2 m, with many steep slopes and flat depressions in the middle, which is a typical karst peak cluster landform. The soil is developed from dolomite and limestone. The infiltration capacity of the soil is stronger, and the inorganic carbon content is higher, resulting in a weak alkaline soil. A large amount of bare rock is exposed in depressions and slopes. The bare rock rate ranges from 15% to 30%. The surface land is mostly covered with gravel, and the volume content of surface soil gravel is between 10% and 40%.

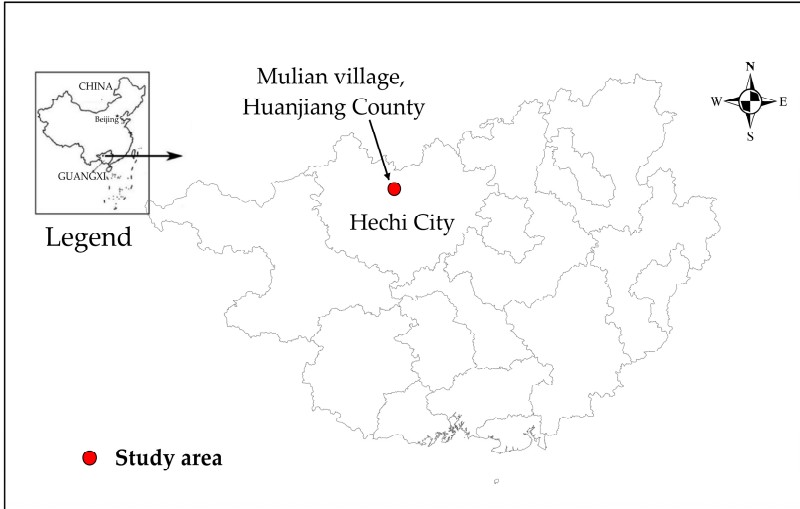

**Figure 1.** Study area bitmap.

Huanjiang County has a subtropical monsoon climate, with an average annual temperature of 19.9 °C and an average annual rainfall of 1400 mm. The rainfall is abundant, but the time heterogeneity is large. The rainy season is concentrated from April to September and accounts for more than 70% of the annual rainfall, of which floods often occur in July, while the dry season (October–March of the following year) has little rainfall, and the monthly average rainfall is less than 90 mm, which is prone to drought. According

to on-site weather station monitoring, the number of extreme weather occurrences has increased significantly increased in recent years [37].

### 2.2. Experimental Design

### 2.2.1. Soil Trough

The adjustable slope steel soil trough was independently designed and movable, with a length of 2 m, a width of 1 m, and a depth of 60 cm (Figure 2). The steel soil trough was 50 cm from the ground, and the slope was freely adjustable from 0 to 30°. The lower end of the soil trough was V-shaped and was connected with the collecting bucket through a hose to collect the runoff and sediment generated on the surface. A small hole with a diameter of 2 cm was reserved on the left side of the lower end of the soil trough to connect the hose and collect the flow from the soil. Six holes were evenly drilled on the bottom plate, with a diameter of 5 cm, of which the effective aperture was 3.5 cm. Underground hole fissure-flow collecting grooves were set under the steel soil grooves. Plastic buckets were used under the water outlet of the collecting groove to collect runoff and sediment samples, and the cumulative runoff and sediment yields were measured after the rainfall.

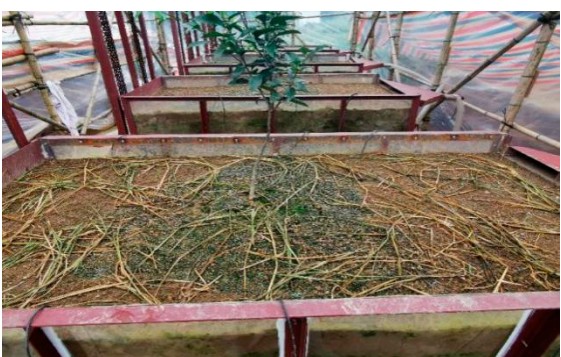

**Figure 2.** Overview of citrus tree soil trough.

### 2.2.2. Rainfall Simulator

The rainfall simulation equipment adopted a "liftable portable simulation rainfall device suitable for high heterogeneity karst slopes". The rainfall simulator consisted of a reservoir and four American SPRACO cone sprinklers, and the rainfall intensity could be controlled by adjusting the cone-type sprinklers.

The height of the rainfall simulator is adjustable, and the height of the rainfall simulator in this study was 4.75 m. When the conical nozzle opening was upward and the water pressure gauge was adjusted to 0.08 Mpa, the end speed of the raindrops obtained during the simulated rainfall was most in line with the natural rainfall characteristics.

### 2.2.3. Test Soil

The soils used in the experiment were all collected from the same small watershed. Among them, the lime soil was collected from yellow lime soil in the top 0–40 cm of the foot of Bande Township, Huanjiang Maonan Autonomous County, Guangxi Zhuang Autonomous Region (24°45′ N, 108°17′ E). The original land-use type of the tested soil was abandoned land. The red soil used in the experiment was collected from the red soil at the surface 0–50 cm of muliantun (24°43′ N, 108°19 E), Huanjiang Maonan Autonomous County, Guangxi Zhuang Autonomous Region. The original land-use type of the tested soil was a citrus garden. After natural air-drying, the test soil was screened for 1 cm, and stored for standby. The basic physical properties of soil are shown in Table 1.

**Table 1.** Basic physical properties of the tested soil.

|  | Soil Bulk Density/(g·cm$^{-3}$) | Clay Content % (0–0.002 mm) | Silt Content % (0.002–0.02 mm) | Sand Content % (0.02–2000 mm) | Organic Matter Content/(g·kg$^{-1}$) |
|---|---|---|---|---|---|
| Lime soil | 1.22 | 64.29 | 24.63 | 10.71 | 12.06 |
| Red soil | 1.36 | 9.86 | 66.53 | 23.61 | 17.00 |

To prevent soil leakage from the bottom of the soil trough during the filling process, the six circular holes at the bottom of the soil trough were covered with water-soluble fibers and then filled with soil. The total thickness of lime soil and red soil was 30 cm and 50 cm, respectively. To ensure the uniformity of soil in the trough, the soil layer was compacted with a self-made iron plate. To reduce the boundary effects, the filling boundary was also compacted.

The simulated bare rock used in the test was dolomite sampled from Xia'nan Township, Hechi City, and the block stone roughly maintains the shape of the bare rock. According to the classification standard of karst rocky desertification, a 10%–13% bare rock rate was simulated in a young lime-soil citrus orchard to represent potential rocky desertification.

### 2.2.4. Tested Citrus Tree and Straw

The citrus plants selected in this study were the most widely planted variety (Orha) in the region. The citrus plants were one year old and the initial tree height was 27.5 ± 3.2 cm. After the citrus seedlings were transplanted into the test soil trough, the simulated rainfall experiment was carried out after 3 months of soil settlement.

To ensure the uniform laying of rice straw, the rice straw was removed from the panicle and evenly cut into small sections of about 15 cm. Before the test, the rice straw was soaked in water for 24 h to ensure it reached the water saturation state and was prevented from absorbing water during the test and affecting the test flow time and flow results. According to the common coverage of local young citrus orchards, the straw coverage area was expanded to a square area of 20 cm around citrus trees. The straw on the soil surface was evenly covered and adjusted to make the covering thickness consistent. The relationship between the coverage of straw mulch and the straw weight can be determined by the formula:

$$MR = -\ln(1 - MC)/Am \tag{1}$$

MR: weight (t/ha); MC: coverage (%); Am: coefficient 0.38.

Therefore, in this observation area, the weight of straw corresponding to 0, 20%, 50%, and 80% coverage was 0, 150 g, 400 g, and 900 g, respectively.

### 2.3. Simulated Rainfall Experiment

The simulated rainfall experiment was carried out from July to September 2020. Four rice straw coverages were set: 0%, 20%, 50%, and 80%. According to the field investigation, the slope of the experimental design was selected to be 5°. Moderate-(60 mm/h) and heavy (120 mm/h)-intensity rainstorms were simulated, and each simulated rainfall was 90 mm, with a total duration of 90 min and 45 min for the above rainfall intensities, respectively [38]. Three repeated tests were carried out for each straw coverage, with a total of 48 effective rainfall events.

During rainfall, surface runoff was collected through a V-shaped collecting trough at the lower end of the trough and then collected in a plastic bucket using a rubber tube. The subsurface flow flows out through holes of 2 cm in diameter at the lower end of the trough and is then collected in plastic drums through a rubber manifold. Groundwater flow was collected through six 5 cm-diameter holes at the bottom of the trough and finally passed through rubber tubes into a plastic bucket. The samples (surface runoff, subsurface flow, groundwater flow) were received in plastic buckets at the same time, and the total weight was obtained by immediate weighing. After all the sediment samples were gathered, the

supernatant was poured and dried in an oven (105 °C) for 24 h, and the sediment weight was obtained by weighing with a balance.

## 3. Results

### 3.1. Reduction in Runoff by Straw Mulching

#### 3.1.1. Effects of Straw Mulching on Runoff Coefficient

The effects of different straw coverages on the surface runoff coefficient, subsurface flow coefficient, and groundwater flow coefficient in lime soil are shown in Table 2. Under moderate rain intensity (60 mm/h), straw mulching obviously reduced the surface runoff coefficient. The surface runoff coefficient of no-straw-covered soil was 2.1, 1.9, and 2.4 times higher than that of straw coverage, at 20%, 50%, and 80%, respectively. The surface runoff coefficient was not significant in the three straw coverages. Compared with the no-straw-covered young citrus orchard, straw mulching significantly increased the subsurface flow coefficient and groundwater flow coefficient of lime soil. The subsurface flow coefficient of 20%, 50%, and 80% straw coverage was 1.5, 1.9, and 1.3 times that of no straw coverage, respectively, and the groundwater flow coefficient was 1.03, 1.1, and 1.4 times that of no straw coverage, respectively. Under heavy rainfall (120 mm/h), straw mulching can still reduce the surface runoff coefficient of lime soil, and the coefficient decreased with the increase in straw coverage. The surface runoff coefficient of no-straw-covered citrus orchards is twice that of 80% straw coverage. A straw coverage of 80% led to a significantly higher groundwater flow coefficient than other treatments.

**Table 2.** Runoff coefficient of red soil and lime soil with different straw coverages.

| Rain Intensity/(mm·h⁻¹) | Coverage/(%) | Red Soil | | | Lime Soil | | |
|---|---|---|---|---|---|---|---|
| | | Surface Runoff | Subsurface Flow | Groundwater Flow | Surface Runoff | Subsurface Flow | Groundwater Flow |
| 60 | 0 | $0.62 \pm 0.01$ a | $0.02 \pm 0.01$ ab | $0.01 \pm 0.001$ a | $0.17 \pm 0.03$ a | $0.08 \pm 0.02$ c | $0.27 \pm 0.09$ b |
| | 20 | $0.64 \pm 0.12$ a | $0.02 \pm 0.01$ b | $0.02 \pm 0.01$ a | $0.08 \pm 0.01$ bc | $0.12 \pm 0.03$ abc | $0.28 \pm 0.05$ b |
| | 50 | $0.60 \pm 0.06$ a | $0.02 \pm 0.01$ ab | $0.02 \pm 0.01$ a | $0.09 \pm 0.01$ b | $0.15 \pm 0.01$ a | $0.29 \pm 0.02$ ab |
| | 80 | $0.56 \pm 0.06$ a | $0.03 \pm 0.01$ a | $0.01 \pm 0.001$ a | $0.07 \pm 0.01$ bc | $0.10 \pm 0.01$ bc | $0.38 \pm 0.005$ a |
| 120 | 0 | $0.43 \pm 0.07$ a | $0.004 \pm 0.001$ a | $0.01 \pm 0.003$ a | $0.26 \pm 0.05$ a | $0.02 \pm 0.01$ b | $0.06 \pm 0.01$ b |
| | 20 | $0.41 \pm 0.09$ a | $0.02 \pm 0.003$ a | $0.004 \pm 0.004$ a | $0.19 \pm 0.02$ b | $0.05 \pm 0.004$ a | $0.07 \pm 0.01$ b |
| | 50 | $0.45 \pm 0.07$ a | $0.004 \pm 0.001$ a | $0.004 \pm 0.002$ a | $0.19 \pm 0.03$ bc | $0.05 \pm 0.01$ a | $0.09 \pm 0.01$ b |
| | 80 | $0.45 \pm 0.03$ a | $0.004 \pm 0.001$ a | $0.004 \pm 0.004$ a | $0.13 \pm 0.02$ c | $0.04 \pm 0.01$ a | $0.14 \pm 0.02$ a |

Different letters in the same column denote significant differences at $p < 0.05$.

However, straw mulching had no significant effect on the surface runoff coefficient, subsurface flow coefficient, and groundwater flow coefficient of young citrus orchards in red soil.

#### 3.1.2. Impact of Straw Covering on Surface Runoff Reduction Rate

After straw mulching, the surface runoff reduction rate of lime soil was significantly higher than that of red soil (Figure 3). At 60 mm/h rainfall intensity, the surface runoff reduction rates of young lime-soil citrus orchards under different straw coverage rates were 49.2%, 41.6%, and 51.5%, and the surface runoff reduction rates of young red-soil citrus orchards were −3.4%, 3.7%, and 10.3%. At 20% straw coverage, the surface runoff was reduced in lime soil, while it was increased in red soil. At 50% and 80% straw coverage, the surface runoff of both lime soil and red soil decreased, but the surface runoff reduction rate of lime soil was 11.2 and 5 times that of red soil. Under heavy rain (120 mm/h), the reduction rates of 20%, 50%, and 80% straw coverage in lime soil were 23.4%, 25.7%, and 49.1%, respectively, and those of red soil were 2.3%, −6.9%, and −5.4%, respectively. With the increase in straw coverage, the reduction rate of lime soil gradually increased. However, in the red soil, only 20% straw coverage had a reduction effect, and the surface runoff was increased at 50% and 80% straw coverage.

*3.2. Effects of Straw Mulching on Surface Erosion Reduction*

3.2.1. Impact of Straw Cover on Surface Erosion

Straw mulching can effectively reduce surface erosion on red-soil slopes, and the total soil erosion is inversely proportional to the straw coverage (Table 3). At a moderate rain intensity (60 mm/h), the surface erosion of red soil without straw coverage was 1504.6 kg/ha, and the reduction rates were 39.3%, 8.5%, and 2.1% for 20%, 50%, and 80% straw coverage, respectively. Under a heavy rain intensity (120 mm/h), the surface erosion was 409.1 kg/ha with no straw coverage, and the surface erosion reached 96.2%, 75.3%, and 24.4% when the straw cover was 20%, 50%, and 80%, respectively. The surface erosion under a moderate rain intensity without straw coverage was 3.7 times as much as that under heavy rain, and it was 1.5, 0.4, and 0.3 times as much under 20%, 50%, and 80% straw coverage, respectively.

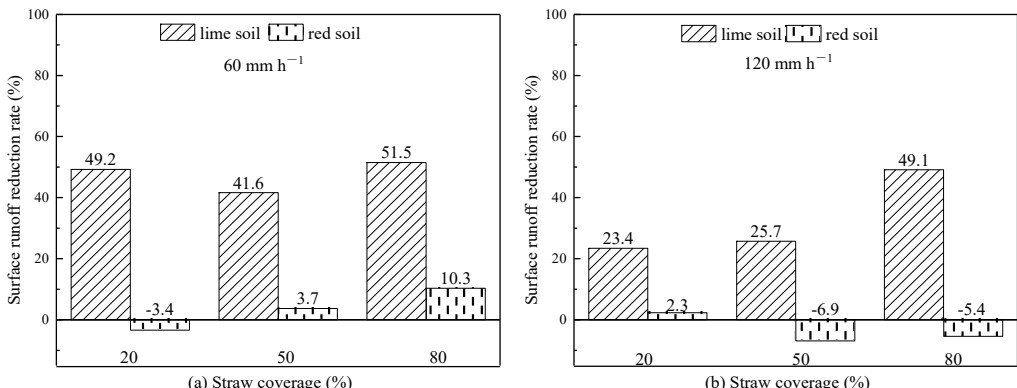

**Figure 3.** Surface runoff reduction rate of red soil and lime soil at 60 mm/h (**a**) and 120 mm/h (**b**) rainfall intensities with different straw coverages. (The surface runoff reduction rate refers to the ratio of the difference between the total surface runoff without straw mulch, minus the total surface runoff with straw mulch, to the total surface runoff without straw mulch).

**Table 3.** Surface erosion of red soil and lime soil with different straw coverages.

| Rain Intensity/(mm·h$^{-1}$) | Soil Type | Straw Coverage | | | |
|---|---|---|---|---|---|
| | | 0 /(kg·ha$^{-1}$) | 20% /(kg·ha$^{-1}$) | 50% /(kg·ha$^{-1}$) | 80% /(kg·ha$^{-1}$) |
| 60 | Red soil | 1504.6 ± 285.3 a | 591.5 ± 79.7 b | 128.3 ± 71.4 c | 32.2 ± 17.7 c |
| | Lime soil | 36.24 ± 13.27 a | 9.69 ± 2.15 b | 6.52 ± 1.27 b | 3.28 ± 0.62 b |
| 120 | Red soil | 409.1 ± 322.8 a | 393.6 ± 528.8 a | 308.2 ± 60.5 a | 100 ± 42.7 a |
| | Lime soil | 212.21 ± 54.69 a | 87.83 ± 11.75 b | 132.67 ± 22.28 b | 11.34 ± 4.88 c |

Different letters in the same line denote significant differences at *p* < 0.05.

At a moderate rain intensity (60 mm/h), the surface erosion on a lime-soil slope without cover measures was 36.24 kg/ha, and the total erosion under 20%, 50%, and 80% straw coverage reduced to 27%, 18%, and 9% of that of no straw coverage, respectively. The results showed that straw mulching can effectively reduce surface soil loss, and the greater the straw coverage, the lower the surface soil loss. Under heavy rain conditions (120 mm/h), the surface erosion of lime soil without straw cover measures was 212.21 kg/ha, and that of 20%, 50%, and 80% straw coverage were 41%, 63%, and 5% of that without straw coverage. Straw cover measures can significantly reduce surface erosion. Under the condition of straw coverage, there is a significant difference between the surface erosion of each cover, and the surface erosion is inversely proportional to the percentage of straw coverage.

3.2.2. Impact of Straw Coverage on Surface Sediment Reduction Rate

Under a moderate rain intensity (60 mm/h), the sediment reduction rates of lime soil and red soil were proportional to the straw coverage (Figure 4). The sediment reduction

rates of 20%, 50%, and 80% straw coverage in lime soil were 71.6%, 80.7%, and 90.5%, respectively, and those in red soil were 60.7%, 91.5%, and 97.9%, respectively. At 20% straw coverage, lime soil had a better sediment reduction effect, and its sediment reduction rate was 1.2 times that of red soil. At 50% and 80% straw coverage, the sediment reduction effects of red soil were 1.1 times higher than those of lime soil.

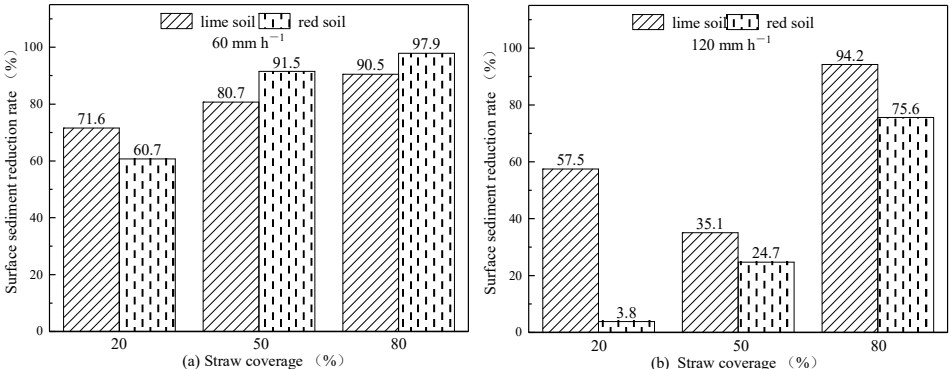

**Figure 4.** The Sediment reduction rate of red soil and lime soil at 60 mm/h (**a**) and 120 mm/h (**b**) rainfall intensities with different straw coverage. (The sediment reduction rate refers to the ratio of the difference between the total surface sediment production without straw mulch, minus the total surface sediment production with straw mulch, to the total surface sediment production without straw mulch).

Under heavy rain conditions (120 mm/h), the sediment reduction rates of 20%, 50%, and 80% straw coverage in lime soil were 57.5%, 35.1%, and 94.2%, respectively, and those in red soil were 3.8%, 24.7%, and 75.6%, respectively. The sediment reduction rates of lime soil were significantly higher than those in red soil, which were 15.1, 1.4, and 1.2 times higher for 20%, 50%, and 80% straw coverage, respectively. With the increase in straw coverage, the difference in the sediment reduction rates of lime soil and red soil decreased.

### 3.2.3. Effects of Straw Mulching on the Sediment Carrying Capacity of Surface Runoff

To analyze the influence of runoff separation and transport capacity on slope erosion, it is reasonable to compare the sediment rate under the same runoff conditions on different surfaces [39].

The straw-covering measures can significantly reduce the sediment-carrying capacity of surface runoff in lime soil (Figure 5). At a moderate rain intensity (60 mm/h), the scatter points without a straw cover extended from the lower-left corner to the upper-right corner, and the erosion rate was proportional to the intensity of surface runoff. However, with straw coverage, the scatter point distribution was mainly concentrated in the lower-left corner, and the maximum runoff intensity was only about 7.5 mm/h. With the increase in runoff intensity, the erosion rate had no significant increase, and the surface soil loss rate of 20% and 50% straw coverage had a slightly decreasing trend with the increase in runoff intensity.

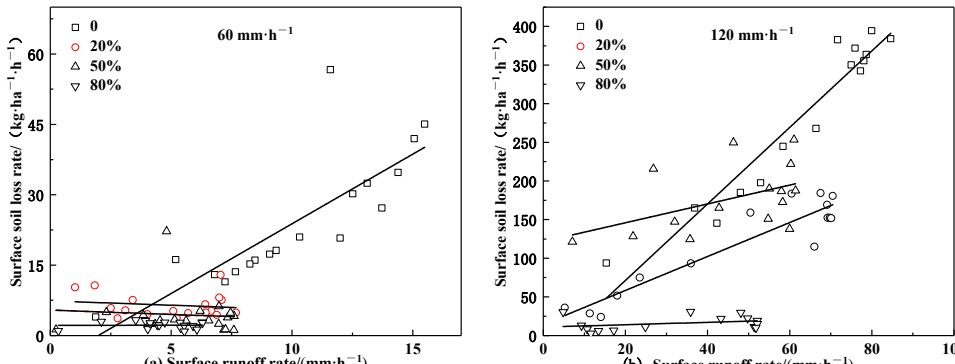

**Figure 5.** Relationship between surface runoff rate and surface soil loss rate in young lime soil citrus orchards at 60 mm/h (**a**) and 120 mm/h (**b**) rainfall intensities under different straw coverages.

Under the conditions of heavy rain intensity (120 mm/h), the distribution of scatter points without straw coverage developed towards the upper-right corner, and when the rain intensity reached the stable state, the scatter point distribution was relatively concentrated. The slope of the scatter points' fitting line without straw coverage at a heavy rain intensity was higher compared with a moderate rain intensity. The scatter distribution at 20% and 50% straw coverage tended towards the upper-right corner, and their surface runoff rate and surface soil loss rate were significantly lower than those without straw coverage. The scatter points at 80% straw coverage tended towards the right, the surface runoff rate and surface soil loss rate were both the smallest, and the surface soil loss rate did not increase with runoff intensity. The results show that straw mulching effectively reduced the sediment-carrying capacity of the surface runoff of lime soil, and the sediment-carrying capacity of runoff decreased with the increase in straw coverage.

The influence of different straw coverages on the runoff and sediment-carrying capacity of a red-soil slope is shown in Figure 6. At a moderate rainfall intensity (60 mm/h), the scatter points without a straw cover are mainly distributed in the upper-right corner. On the one hand, the surface runoff rate increased rapidly and reached a stable runoff state. On the other hand, the surface soil loss rate increased with runoff intensity. The scattering characteristics of 20% straw coverage were similar to those of no straw coverage, both of which increased with the surface runoff rate. From the distribution of scatter points, the maximum surface runoff rate of 20% straw coverage exceeded that without straw coverage, while the surface soil loss rate under the same surface runoff rate was significantly lower than that without straw coverage. The scattering characteristics at 50% and 80% straw coverage were similar, and their surface soil loss rates fluctuated with the increase in surface runoff rate. It was concluded that the surface runoff sediment-carrying capacity of red soil is inversely proportional to straw coverage at a moderate rain intensity.

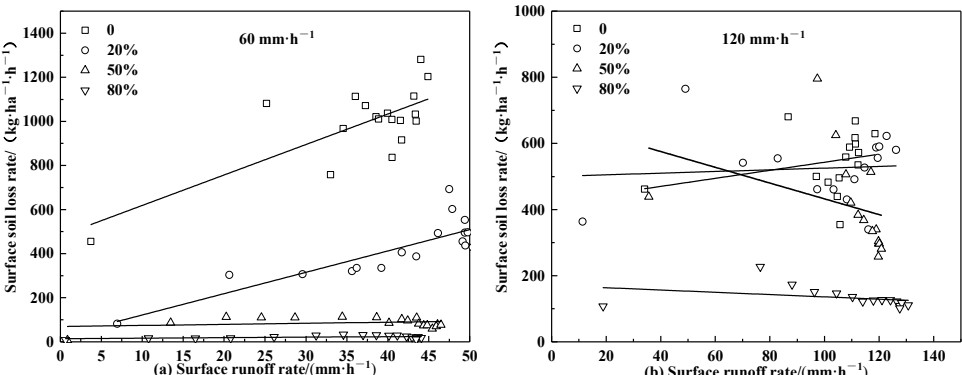

**Figure 6.** Relationship between surface runoff rate and surface soil loss rate in young red soil citrus orchards at 60 mm/h (**a**) and 120 mm/h (**b**) rainfall intensities under different straw coverages.

Under heavy rain conditions (120 mm/h), the scatter points' distribution without straw coverage was mainly concentrated in the upper right corner. In the more concentrated part, the scatter showed a vertical distribution from top to bottom. This may be because, after the local surface runoff reached a stable state, the surface soil rapidly began to crust, and the soil that could be eroded decreased; this, in turn, caused the erosion rate to decrease under a high runoff intensity. The scattering characteristics of the 20% straw cover spread from the lower-left corner to the upper-right corner, and its surface soil loss rate increased slightly with the increase in surface runoff rate. The distribution of scatter points for 50% and 80% straw coverage were mainly concentrated in the lower right corner, and their surface soil loss rate decreased with the increase in runoff intensity. The scatter points' distribution of 80% straw coverage was significantly lower than that of other straw coverages.

## 4. Discussion

Straw mulching can effectively reduce the surface runoff coefficient of young lime-soil citrus orchards and increase the subsurface flow and groundwater flow coefficient at the same time. After straw mulching, the runoff source of young lime-soil citrus orchards changed from the surface to underground. Under the same rainfall intensity, straw mulching can reduce the direct contact between raindrops and surface soil and reduce the flow rate of surface runoff. Increasing straw coverage will weaken the kinetic energy of raindrops, and improve rainwater infiltration, soil flow, and groundwater flow [40]. The effect of straw mulching on runoff coefficients in red soil was not significant. During rainfall, the impact of raindrops on the exposed surface soil will cause soil surface crusting, reduce soil infiltration, and hinder rainwater infiltration [39]. To ensure that the initial water content is basically the same before each test, precipitation with a rainfall intensity of 30 mm/h was carried out 24 h before the simulated rainfall test. Studies have shown that the stable infiltration rate of soil in the karst area of the study area is 42.0–255.0 mm/h. In this study, 30 mm/h rainfall intensity was used for pre-wetting. The stable infiltration rate of the soil in the study area is much higher than the rainfall intensity for pre-wetting, so surface runoff will not occur during pre-wetting [41]. In the formal simulated rainfall test, the soil moisture basically reached the saturated state, the infiltration rate of the red soil in the karst area was low, and the infiltration decreased with the increase in previous water content. During rainfall, the surface runoff is produced quickly due to the lower soil infiltration rate and greater rain intensity. Rainwater can easily generate runoff directly onto the straw, rather than produce surface runoff after contact with the surface soil, so its impact on the runoff coefficient is not significant [42]. McDowell's results showed that a higher early soil water content will increase surface runoff [43].

Straw mulching can significantly reduce surface erosion. The surface erosion decreased with the increase in straw coverage, which is consistent with most of the research results [44–46]. In this study, straw mulching increased the roughness of the soil surface and reduced sediment transport, while surface runoff decreased with the increase in straw coverage, further reducing the runoff erosion power. At a moderate rainfall intensity (60 mm/h), the ground erosion of lime soil under different straw coverages was significantly lower than that of red soil. This is because the silt and sand content in red soil is higher than in lime soil. During rainfall, silt and sand in soil are more easily transported by surface runoff, which results in the increased erosion of red soil in young citrus orchards under the same straw coverage. However, under heavy rain intensity (120 mm/h), the surface erosion of a young red-soil citrus orchard without straw coverage is significantly lower than that under moderate rain intensity (60 mm/h). This is due to the stronger impact of heavy-rain-intensity (120 mm/h) raindrops on surface soil without cover, which means that the surface more easily forms a soil crust, and thus reduces surface erosion.

Under heavy rainfall (120 mm/h), the surface erosion of red soil at 20% straw coverage is 393.6 ± 528.8 a. The variability of these experimental data is very high, because, in one of the experiments, the surface soil simulated rainfall. A continuous collapse occurred in the middle of the experiment, and large pieces of surface soil were washed away by surface

runoff, resulting in large differences in the data of the three experiments. This phenomenon shows that, even under the condition of straw mulching, the surface soil of red-soil orange orchards is extremely unstable under heavy rain conditions, and is prone to gully erosion, which may cause more serious surface soil loss.

Based on the analysis of the relationship between runoff intensity and erosion rate, the runoff sediment-carrying capacity of young lime-soil citrus orchards without straw coverage was higher than that with straw coverage. This is because a straw cover reduced the surface runoff; therefore, the soil erosion power was insufficient to transport sediment during the rainfall–runoff process. After surface crusting occurred at the later stage of rainfall, the possibility of soil particles being peeled off from the surface also decreased. Therefore, under the condition of straw coverage, the dual effects of limited runoff transportation and limited soil peeling are formed.

Under moderate rain intensity (60 mm/h), the slope of the fitting line between the surface runoff and the sediment-carrying capacity of red soil decreased with the increase in straw coverage. The scatter points of uncovered measures are mainly concentrated in the upper right, and a higher runoff intensity was reached most quickly, which indicated that erosion was mainly caused by runoff scouring. However, after straw mulching, although the runoff intensity did not decrease significantly, its erosion rate decreased significantly, because straw mulching reduced the runoff speed in direct contact with soil to a certain extent, and straw also intercepted a large amount of sediment. Miyata's research on Japanese cypress plantations showed that, in covered plots, floor coverage prevented 95% of soil detachment by raindrops, which was the dominant mechanism in reducing soil erosion, similar to our research results [47]. Under heavy rain (120 mm/h), the erosion rate of 80% straw coverage was the lowest compared with 0%, 20%, and 50% straw coverage, and the erosion rate decreased with the increase in runoff intensity. With higher rain intensity and straw coverage, precipitation was more likely to produce runoff directly on the straw surface, while the part in direct contact with the soil surface has a lower flow rate, and the soil surface more easily forms a crust. Straw also has a stronger retention effect on soil, thus reducing the possibility of soil erosion.

Finally, the results of our study were obtained through an artificial rainfall test simulating a rainstorm. To better study the runoff and sediment production characteristics of young orange orchards covered with straw in karst areas of Southwest China, it is necessary to further observe and study the runoff and sediment yield characteristics of young orange orchards through natural rainfall and further explore the loss of soil nutrients during rainfall.

## 5. Conclusions

In our study, we used simulated rainfall experiments to study the soil and water conservation benefits of young red-soil and lime-soil citrus orchards under different straw coverages. The main conclusions are as follows: straw mulching can significantly reduce surface erosion in both lime soil and red soil in young citrus orchards. The results proved the first hypothesis. In lime soil, the surface sediment reduction rate after straw mulching can reach 35.1%–94.2%. In red soil, the surface sediment reduction rate with more than 50% straw coverage can achieve a significant surface erosion reduction effect of 24.7%–97.9%. The effects of straw mulching on reducing surface runoff in lime soil were significantly higher than in red soil, which verified the second hypothesis. The surface runoff reduction rate of the lime soil after straw mulching can reach 23.4%–51.5%, but for red soil, the rate was only −6.9%–10.3%. The main reason for the reduction in surface erosion of lime soil by straw mulching is limited runoff transportation, while for red soil, the main reason is the decrease in soil stripping.

The soil and water conservation effect of 80% straw coverage is the best, but too high of a straw coverage rate will lead to adverse effects on agricultural machinery. Therefore, 50% straw coverage is recommended, which is enough to provide a similar soil and water conservation effect for young lime-soil and red-soil citrus orchards.

**Author Contributions:** Conceptualized this study, Z.G. and Q.X.; designed the study, Z.G. and Q.X.; performed data checks, S.Z., Z.F. and H.C.; wrote the original draft preparation, Z.G. and Q.X.; edited and revised the original manuscript, Z.G. and Q.S. All authors have read and agreed to the published version of the manuscript.

**Funding:** This research was funded by the Guangxi Natural Science Foundation of China (2020GXNS-FAA297093 and 2018GXNSFGA281003) and the National Natural Science Foundation of China (NO.51769005).

**Institutional Review Board Statement:** Not applicable.

**Informed Consent Statement:** Not applicable.

**Data Availability Statement:** The data are available upon reasonable request by contacting the corresponding author.

**Acknowledgments:** The author would like to thank Guilin University of Technology and The Institute of Subtropical Agriculture, The Chinese Academy of Sciences for their assistance. Moreover, we also thank the editors and anonymous reviewers for their review of the article and their insightful and constructive comments.

**Conflicts of Interest:** The authors declare no conflict of interest.

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
