# Peer review of "Effects of Different Straw Mulch Rates on the Runoff and Sediment Yield of Young Citrus Orchards with Lime Soil and Red Soil under Simulated Rainfall Conditions in Southwest China"

_water, doi:10.3390/w14071119_

Round 1
Reviewer 1 Report
I had the chance to review this article on its submission to Sustainability and Water. Compared with that submission, this version is improved, but not enough.
The subject of the article is interesting, and it is linked to the objectives of the journal, however, there are some issues that have to be reconsidered.
For better visibility on databases, the authors are asked not to repeat among keywords the words/concepts included in the title of the article.
The results are interesting and they are well discussed, but the conclusions are not enough to sustain the results. The use of the research is, so, insufficiently explained at the conclusion part, it seems just mention again the results. It is advisable to use that part for formulating general conclusions and recommendations for scholars, government, business etc..
Author Response
Response to Reviewer 1 Comments
Point 1: For better visibility on databases, the authors are asked not to repeat among keywords the words/concepts included in the title of the article.
Response 1: Thank you for your suggested revisions to the manuscript. As you mentioned, the key words in the manuscript are repeated with some words in the title. To avoid this problem, we have reconsidered the choice of key words. After revision, we have changed the keywords in the revised manuscript to: "simulated rainfall; soil erosion; orchard; conservation tillage; ecological fragile zone."
Point 2: The results are interesting and they are well discussed, but the conclusions are not enough to sustain the results. The use of the research is, so, insufficiently explained at the conclusion part, it seems just mention again the results. It is advisable to use that part for formulating general conclusions and recommendations for scholars, government, business etc.
Response 2: Thank you for your suggested revisions to the manuscript. Indeed, we ignored this point before, and our conclusion is to provide the local fruit growers and the government with the optimal straw mulch coverage to achieve good soil and water conservation in lime soil and red soil orchards. Therefore, we made some supplements in the conclusion part, and the supplementary content is: "The soil and water conservation effect of 80% straw coverage is the best, but too high straw coverage will have some adverse effects on agricultural machinery. Therefore, it is recommended to use 50% straw coverage, which is enough to provide similar soil and water conservation effect for lime soil and red soil young citrus orchards." Thank you again for your suggestions.

Reviewer 2 Report
An English-language proofreading of the entire text by a native English speaker is recommended.
Authors should be more careful when presenting the results.
The comments can be found in the attached pdf.

Author Response
Response to Reviewer 2 Comments
Point 1: An English-language proofreading of the entire text by a native English speaker is recommended.
Authors should be more careful when presenting the results.
Response 1: Thank the reviewers for their suggestions on the English expression of the manuscript. After checking the full text of the manuscript by all authors, we found some inappropriate expressions in English. Therefore, after this modification, we will seek the'MDPI'English editing service to modify the revised draft in English to avoid problems in English. On behalf of all the authors, I would like to thank you again for your proposed amendments.
Point 2: keywords are repeated in the title, choose others
Response 2: Thank you for your suggested revisions to the manuscript. As you mentioned, the key words in the manuscript are repeated with some words in the title. To avoid this problem, we have reconsidered the choice of key words. After revision, we have changed the keywords in the revised manuscript to: "simulated rainfall; soil erosion; orchard; conservation tillage; ecological fragile zone."
Point 3: -“Agronomic measures such as straw mulching and returning to field are typical
protective tillage measures.”
-”check english”
Response 3: Thank you for your suggestions for revision of the draft. As you mentioned, this sentence really doesn't express clearly, and we have re-described it in the revised version with the following specific modifications:'Straw mulching has been proven to be a typical conservation tillage measure'.
Point 4: -“Rahma's research showed that straw mulch can induce greater soil losses from loess slopes than no mulch under extreme rainfall conditions [33]. Somchai's research on sandy loam soil in northern Thailand showed that straw mulch can effectively reduce surface runoff and surface soil loss. Among them, a straw mulch of 5.0 t/hm² has the best soil and water conservation benefits [34].”
-”it seems ideas that are contraditory. Clarify”
Response 4: Thank you for your suggestions on the revision of the manuscript. The research results of "rahma" and "Somchai" mentioned in the manuscript are opposite, but they are not contradictory. We are in "rahma's research shown... Has the best soil and water conservation benefits [34]." The previous sentence indicates a view that straw mulching has different water and soil conservation effects in different soil types, and even straw mulching will aggravate the risk of water and soil loss in some types of soil. Later, the opposite research results of "rahma" and "Somchai" using straw mulching in different soil types are cited to prove this view, Our research is also to compare the soil and water conservation effects of straw mulching on two soil types, so there is no contradiction between them.
In order to better explain the difference of soil and water conservation effect of straw mulching on different soil types, we further explained the reasons for this difference in the "Introduction". Thank you again for your suggestions.
Point 5: Soils should be well characterized in terms of composition, texture, structure, CTC, pH, water storage capacity and wilting coefficient, water infiltration characteristic, depth, characterization of soil horizons.
Response 5:Thank you for your suggested revisions to the manuscript. Since our study tested both red soil and lime soil, we described the characteristics of the two soils, such as soil thickness, composition, in as much detail as possible in '2.2.2' and 'Table 1'. However, our study requires further processing of the experimental soil, so we did not conduct more abundant tests on the soil, such as pH, water storage capacity, water infiltration characteristics, etc. However, in the following research, we will follow your suggestion to devote more attention to the study of these properties of soil. Thank you again for your suggestion to revise the manuscript.
Point 6: -”I think is a low deep of soil for citrus tree”
Response 6: Thank you for your questions about the experimental setup in the manuscript. The study area belongs to the karst area in Southwest China, especially in Huanjiang County, with typical karst geomorphic characteristics. Before the experiment, we conducted a field investigation on the local orchards. The soil layer of citrus orchards on lime soil slope is relatively shallow, only about 30cm, and the soil layer thickness of citrus orchards on red soil slope is about 50cm. Therefore, these two typical soil layer thicknesses were simulated in our study.
Point 7: -‘When filling the soil tank with experimental soil, in order to prevent the experimental soil from leaking from the six holes at the bottom of the soil tank, cover the six holes with water-soluble fibers with a side length of 10cm square, and then fill the soil.’
-’Check english’
Response 7: Thank you for your suggestion to revise the English of the manuscript. After our verification, this sentence is indeed not clearly expressed, and we have re-described this sentence in the revised manuscript.
Point 8: -’The citrus tree variety used in the experiment was 1-year-old citrus seedlings in Northwest Guangxi, China. and the initial tree height was 27.5 ± 3.2 cm.’
-’what is the variety?’
Response 8: Thank you for your questions and suggestions for revisions to the manuscript. There are many citrus orchards in the study area, and the citrus varieties grown in each orchard are different, but the plant variety we selected in the experiment is "Orah", which is the most widely planted in the area. We have re-described the reasons for the selection of plant varieties in the revised manuscript. Thank you again for your suggested revisions to the manuscript.
Point 9:
-MR=-ln(1-MC)/Am |
(1) |
MR: weight (t / ha), MC: coverage (%), Am: coefficient 0.38.
-’describe each abreviation in the text above the expession .
What is the values of MR and MC used in this expression to obtain am value of 0.38.
What is the meaning of am?
The symbol of tonnes is t instead T’.
Response 9: Thank you for your suggested revisions to the manuscript. We have refined the description of this formula in the revised manuscript as you suggested. "MR" refers to the weight of straw that should be laid evenly per hectare of land (unit: t/ha), "MC" refers to the target laying straw coverage (range: 0-1), "Am" is a formula of the A fixed coefficient in , with a coefficient of 0.38.We can calculate how much straw needs to be laid under this straw coverage through "MR" and the land area.
In this study, the simulated straw coverage (MC) is 0,20%, 50%, 80%, and the obtained MR is 0,0.59,1.82,4.24.Finally, the weight of straw we need to cover is obtained by unit conversion and rounding.
Point 10: What do the authors mean with interflow?
Response 10: Thank you for your questions about the manuscript. We have carefully considered this word problem. Finally, we found the correct academic term. Now we have changed "interflow" to "subsurface flow" in the full text. 'Subsurface flow' refers to the lateral flow of water at the interface of the soil surface or within the layered soil, also known as surface flow. is part of the runoff. Thank you again for your questions.
Point 11: ‘check the value presented in table 3, it was 212.21’
Response 11: Thank you for the error you made in the manuscript. It is indeed our negligence. After careful review, we determined that the data in Table 3 is correct. We mistakenly wrote 212.21 to 117.08 in '3.2.1'. We regret this major mistake. After our review, we have corrected the error, checked the data in the full text of the manuscript, and corrected the errors in the full text. The errors in the manuscript you raised are of great importance to us, and we would like to express our sincere thanks again.
Point 12: ‘kg without capital letter, check all the text’.
Response 12: Thank you for pointing out a typo in the manuscript, which we have corrected throughout the text and will change 'kg' to lowercase in the revised manuscript.

Reviewer 3 Report
The lime soil sample was taken from abandoned land - is that representative to soil properties of the citrus gardens (that is the study focus)? Would not the soil properties be different if taken form real garden (especially Corg)?
Why the sample thickness was different for the two soil types? At 30 cm the percolation still have to be different from natural soil conditions - so the infiltration and runoff coefficient have to be hardly interpretable, ...
Or is it representing typical soil profile depths for both soil types in the region, and is the soil sample with bedrock always 60 cm as indicated in first paragraphs? The setup is a bit unclear. Is the infiltration properties of the bedrock kept "natural" or typical? How that was reached?
Were the samples compacted to reach the original bulk density? Only by iron plate - or was there any water saturation experiment to get closer to original soil properties? If not waiting for consolidation, the experiment is hardly comparable to natural situation. Or was that established by the 3 months period of waiting?
Tree months of the plants and samples to get to good status looks fine - until that it was under natural rainfalls only? what was the rain amount it got before the experiment - was that normal for the area?
Was the preparation of the straw typical for the region - is it being cut similarly (15 cm) in nature?
What was the reason to rain in 60 and 120 mm/h intensities?
You mention that the soil slope is adjustable - but only 5° inclination was tested - there are no conclusions concerning potential effects of the slope to the differences between the soils presented. Some literature could provide opinions on that.
There is no discussion on the variability of the replications. From table 2 it is visible it was not too high for runoff, but there is no explanation if the proposed trends are significant, and if the numbers of experiments is sufficient to formulate answers. Since from table 3 it is clear that variability is very high for sediment transports - that is typical for all RS experiments especially on smaller scale (look at 120 mm/h, 20% red soil).
Discussion:
"To ensure that the initial water content is basically the same before each test,precipitation with a rainfall intensity of 30 mm/h is carried out 24 hours before the simulated rainfall test. " - no discussion - this belongs to method section, where it was not stated. And I think 30 mm/h should already lead to surface runoff and crusting - so to my opinion it is too heavy rain for prewetting.
After resolving the raised issues, clarifying the validity of the results and polishing the discussion section, the manuscript is acceptable for publishing.
Author Response
Response to Reviewer 3 Comments
Point 1: The lime soil sample was taken from abandoned land - is that representative to soil properties of the citrus gardens (that is the study focus)? Would not the soil properties be different if taken form real garden (especially Corg)?
Response 1: We would like to thank the reviewers for their scientific questions about the manuscript, which we will explain in detail. In our study, the orchard was simulated manually, and all the soils in the experimental groups were treated the same way. First, we placed the returned soils in sunlight for natural air drying, then passed the naturally air-dried soils through a 10 mm sieve hole size, and finally filled the sieved soils into the trough immediately. As you asked, in our experiment, lime soil was taken from waste land, but no matter what type of land use, there would be any significant difference in soil properties after treatment because the purpose of soil treatment is to control soil bulk better when filling the soil tank.
On the other hand, through interviews with local fruit growers, we know that many new citrus orchards in the area have been converted from waste land to citrus orchards. Therefore, in our study, lime soil samples were taken from waste land to better simulate the field situation of "young orchard" after reclamation.
Point 2: Why the sample thickness was different for the two soil types? At 30 cm the percolation still have to be different from natural soil conditions - so the infiltration and runoff coefficient have to be hardly interpretable, ...
Or is it representing typical soil profile depths for both soil types in the region, and is the soil sample with bedrock always 60 cm as indicated in first paragraphs? The setup is a bit unclear. Is the infiltration properties of the bedrock kept "natural" or typical? How that was reached?
Response 2: We would like to thank the reviewers for their scientific questions about the manuscript, which we will explain in detail. As you mentioned, the sample thicknesses of the two soil types in our study are different. According to our field survey, the typical soil thicknesses of the two soil types in the study area are different. The soil thicknesses of lime soil orchards are usually about 30cm and those of red soil orchards are usually about 50cm. Therefore, our research simulates these two typical soil thicknesses. The soil tank is 60 cm deep and can simulate soil environments with soil depths less than 60 cm. Therefore, the soil tank is suitable for soil thicknesses of 30 cm and 50 cm simulated in our study.
We have partially modified the description of the test settings in the manuscript "2.2. Experimental design".
Point 3: Were the samples compacted to reach the original bulk density? Only by iron plate - or was there any water saturation experiment to get closer to original soil properties? If not waiting for consolidation, the experiment is hardly comparable to natural situation. Or was that established by the 3 months period of waiting?
Response 3: We would like to thank the reviewers for their scientific questions about the manuscript, which we will explain in detail. In this study, we determined the soil bulk density to be simulated by investigating the bulk density of two local soil types of citrus orchards. Since the trough is a regular cuboid, we can calculate the weight of the soil to be filled per unit thickness of the soil by simulating the bulk and thickness of the soil to be simulated. During the filling process of soil tank, we fill the soil once every 10cm, fill the corresponding weight of soil, then compact it to 10cm scale with iron plate to ensure that the typical soil bulk density of citrus orchard of this soil type is simulated.
In order to make the simulated citrus orchard closer to the natural conditions, we set the soil down naturally for three months after planting the seedlings of citrus trees to consolidate the soil and make it better simulate the natural citrus orchard soil environment.
Point 4: Tree months of the plants and samples to get to good status looks fine - until that it was under natural rainfalls only? what was the rain amount it got before the experiment - was that normal for the area?
Response 4: Thank you for your questions about the experimental setup of the manuscript. After transplanting the citrus seedlings into the soil tank, we will carry out natural soil settlement for three months. The natural soil settlement time is from April to June. During this period, we hired local fruit farmers to take care of and maintain the citrus seedlings. The common planting habit of local fruit farmers is to add additional artificial irrigation at the initial stage of Citrus seedling planting, To ensure that citrus seedlings will not wither due to drought in hot weather. Therefore, in this study, citrus seedlings will receive additional watering from fruit farmers while receiving natural rainfall. The amount of additional watering for each citrus seedling is equal, but the amount of watering is controlled by fruit farmers. In June, the rainfall in the study area increased, and all citrus seedlings were only watered by natural rainfall. The formal test was conducted from July to September.
Point 5: Was the preparation of the straw typical for the region - is it being cut similarly (15 cm) in nature?
Response 5: Thank the reviewers for their scientific questions on the manuscript. We will explain this issue in detail. Before the experiment, we interviewed local fruit farmers and asked about the local straw treatment methods. The answer we got was: after the straw is recycled, it will be manually cut and covered in the orchard. The fruit farmers did not give an accurate answer to the straw cutting length, but we measured the length of the straw cut by the fruit farmers interviewed. Most of the straw length after cutting is about 15cm, so we adopted this typical local straw treatment method.
Point 6: What was the reason to rain in 60 and 120 mm/h intensities?
Response 6: Thank the reviewers for their questions about the manuscript and we will explain this in as much detail as possible. Our study mainly simulated the runoff and soil loss characteristics of citrus orchards under rainstorm with straw cover measures. 60 mm/h and 120 mm/h are typical rainfall intensities in the study area. On the other hand, July-September is the most time when rainstorm occurs in the study area every year. The temperature is higher in this period and the rains and heat are in the same period. This is also the reason why we will arrange the simulated rainfall test in July-September.
Point 7: You mention that the soil slope is adjustable - but only 5° inclination was tested - there are no conclusions concerning potential effects of the slope to the differences between the soils presented. Some literature could provide opinions on that.
Response 7: Thank the reviewers for their suggestions on the manuscript. The soil trough gradient used for the test is adjustable, but in this study we only simulated a 5 degree gradient, which is due to our previous research that the local slope of the citrus orchard is mostly about 5 degrees, which is a typical gradient. The local slope less than 5 degrees has been planted with more suitable crops, such as mulberry and corn. It is undeniable that there are citrus orchards with a slope greater than 5 degrees, but the area of the citrus orchard with this slope is much smaller than that of the citrus orchard with a slope of about 5 degrees, because the larger slope increases the difficulty of citrus planting and the agricultural machinery is difficult to use on steep slope. The citrus orchard with a slope of about 5 degrees has a larger planting area, which has become a disaster area for water and soil loss. Therefore, we simulated the most representative citrus orchard with a slope of 5 degrees.
Point 8: There is no discussion on the variability of the replications. From table 2 it is visible it was not too high for runoff, but there is no explanation if the proposed trends are significant, and if the numbers of experiments is sufficient to formulate answers. Since from table 3 it is clear that variability is very high for sediment transports - that is typical for all RS experiments especially on smaller scale (look at 120 mm/h, 20% red soil).
Response 8: Thank you for your suggestions for revision of the draft. Our research data used an average of three repeated trials and when discussing the results, we tended to discuss the trend of data changes due to different straw coverage, thus missing a discussion on whether the data were different. In the study, three repeated tests were sufficient to obtain accurate test data under strict control of test control variables.
As you mentioned in Table 3, the data of soil loss with 20% straw coverage in red soil at 120mm/h rain intensity are quite different. We found in the test records that during a simulated rainfall with this measure, the surface soil collapsed continuously and the agglomerated soil was washed away by surface runoff, resulting in large differences in the data of the three tests. This also indicates that the surface soil of red soil is more prone to soil loss under heavy rain. Even after straw mulching, it may still collect surface runoff due to straw accumulation, resulting in furrow erosion on the surface of soil, causing serious surface soil loss. We have added discussion and clarification about this part in '4. Discussion' section. Your suggestion for this revision is very important to us, and we thank you again.
Point 9: "To ensure that the initial water content is basically the same before each test,precipitation with a rainfall intensity of 30 mm/h is carried out 24 hours before the simulated rainfall test. " - no discussion - this belongs to method section, where it was not stated. And I think 30 mm/h should already lead to surface runoff and crusting - so to my opinion it is too heavy rain for prewetting.
Response 9: Thank the reviewers for their suggestions on the manuscript. Previous studies have shown that the stable soil infiltration rate of slope land in local karst area is 42.0-255.0 mm / h, while the prewetting rainfall intensity in the study is 30 mm / h. The soil infiltration rate in this area is significantly higher than the prewetting rainfall intensity. We conducted a test in the preliminary pre experiment to record the time when the surface runoff starts during prewetting. In the formal experiment, when we conduct prewetting, we will stop simulating rainfall before generating surface runoff to prevent surface runoff during prewetting.
We have reinterpreted this problem in "4 Discussion " and added References: "Chen H.S, Liu J.W, Zhang Wang K. soil hydraulic properties on the steel karst hillslopes in Northwest Guangdong, China. Environment. Earth SCI. 2012,66:371-379."

Reviewer 4 Report
The authors presented the studies on the effect of different rates of straw mulch on runoff and sediment yield on lime and red soil in studies with a rainfall simulator. The studies showed that application of straw mulch was more effective in reduction of sediment on limestone than red soil. The manuscript contains interesting studies that are important for the practice.
However, the manuscript contains a lot unclearness, that are related most probably with weak or even very weak English in some parts. The title and purpose are formulated in improper way. For example, the purpose is not “to determine the effect of runoff and sediment reduction by rice straw mulching of young citrus orchard with different soil types, four straws coverage (0, 20, 50 and 80%) and two rain intensities (60 and 120 mm/h) in the simulated rainfall experiment” but to determine the effect of different rates of straw mulch on runoff and sediment on lime and red soils in studies with a rainfall simulator. There are a lot of similar improper formulation in the manuscript.
In my opinion, the manuscript needs an extensive correction of English. The text is unclear and difficult to understand in too many places. Without a significant improvement of English, the review cannot be done in a proper way.
I suggest to return the manuscript to authors for improvement of English. In some places, the text should be also reconsidered.

Author Response
Response to Reviewer 4 Comments
Point 1: “In my opinion, the manuscript needs an extensive correction of English. The text is unclear and difficult to understand in too many places. Without a significant improvement of English, the review cannot be done in a proper way.”
Response 1: Thank the reviewers for their suggestions on the English expression of the manuscript. After checking the full text of the manuscript by all authors, we found some inappropriate expressions in English. Therefore, after this modification, we will seek the'MDPI'English editing service to modify the revised draft in English to avoid problems in English. On behalf of all the authors, I would like to thank you again for your proposed amendments.
Point 2: “Among them, a straw mulch of 5.0 t/hm² has the best soil and water conservation benefits [34].”
What is a “t/hm2” unit?
Response 2: Thank you for your suggestions on the manuscript. “5.0 t/hm ²” It means that each hectare of land is evenly covered with 5 tons of mulch, which is a coverage unit.
Point 3: -comparison of lime and red soil.
Authors give a geologic origin of lime soil. What about the red soil?
Response 3: Thank you for your suggestions on the revision of the manuscript. We really lack the origin of red soil in the "Introduction". Red soil is formed by the weathering of carbonates or rocks containing other iron-rich aluminum oxides in hot and humid climates. At present, we have added this part to the manuscript.
Point 4: - “The study in order to verify two main hypotheses:…”
The study was performed in order to….
Response 4: Thank you for your suggestion to revise the English wording of the manuscript. As you mentioned, this belongs to our English grammar error when writing the paper, we have revised the English expression here according to your suggestion, thank you again for your revision suggestion.
Point 5: -“underground runoff and soil runoff”
What means “soil runoff”? Usually “surface runoff” is used. What about “interflow” in table 2? Please, uniform the terms (overland flow, interflow, groundwater flow – if you prefer hydrologic terms for flow)
Response 5: Thank you for your suggestions on the revision of the manuscript. We have indeed neglected to unify the terms of the full text. In the full text, the "soil runoff" mentioned in our research should be expressed as ‘subsurface flow’. 'Subsurface flow' refers to the lateral flow of water at the interface of the soil surface or within the layered soil, also known as surface flow. is part of the runoff. We checked the terms of the full text in the revised manuscript and unified the use of terms. Thank you again for your modification suggestions.
Point 6: -“The soil is developed from dolomite and lime soil.”
“lime” or limestone?
Response 6: Thank you for your suggestions on the revision of the manuscript. We are sorry for this kind of writing error in the manuscript. As you mentioned, the correct statement should be “The soil is developed from dolomite and limestone”. In addition, we also checked other writing errors in the manuscript.
Point 7: “citrus orchards of karst areas” - citrus orchards in karst areas“
Response 7: Thank you for your suggestions on the revision of the manuscript. We are sorry for this writing error in the manuscript. We have corrected this English language error in the revised manuscript. In addition, we have also checked other writing errors in the manuscript.
Point 8: “The measure had a higher runoff and sediment reducing in lime soil than that in red soil” -???
Response 8: Thank you for your suggested revisions to the manuscript. As you suggested, this is a typo in the English language, what we want to express is that under the action of straw mulching, the surface runoff reduction rate of lime soil is much higher than that of red soil, and now we have corrected the expression in the text , thanks again for your suggestion to edit.
Point 9: what is “soil stripping”? The term seems to be used in connection with erosion process.
Response 9: Thank you for asking questions about the words in the manuscript. We will explain this word in as much detail as possible. "Soil stripping" refers to that when surface runoff occurs on the soil surface, the surface runoff peels the surface soil particles from the soil surface due to kinetic energy, and the stripped soil particles are transported with the runoff, resulting in water and soil loss. The kinetic energy of surface runoff is an important factor to determine the stripping of soil by surface runoff, but straw mulching will reduce the kinetic energy of surface runoff, and then reduce the possibility of stripping surface soil particles by surface runoff.
Point 10: Page 6: “Reduction of straw mulching on runoff”???? It could rather be: straw mulching and reduction of runoff (or reduction of runoff).
Response 10: Thank you for your suggested revisions to the manuscript. This section describes the effects of straw mulching on runoff in various aspects. As you said, the previous title was not appropriate, so we changed the title of this section to "Reduction of runoff by straw mulching"
Point 11: Page 12: “Straw mulching measures” – “straw mulching” is enough, you can omit “measures”
Response 11: Thank you for your suggestions on the revision of the manuscript. We have corrected similar English language errors in the manuscript and sought professional English editing services. We will pay more attention to this problem in subsequent research and thesis writing. Thank you again for your suggestions.
Point 12: Figure 5:
Title: “Surface runoff and sediment carrying capacity of lime soil young citrus orchard under
different straw coverage”
title is somewhat awkward. What means “sediment carrying capacity of soil”? The figure presents the relations between soil loss and surface runoff rates.
Response 12: Thank you for your suggestions on the revision of the manuscript. Figure 5 shows the relationship between surface runoff rate and soil loss. The scatter points in the figure represent the amount of soil loss at the current surface runoff rate, and the binary primary straight line fitted according to these scatter points represents the ability of surface runoff to cause soil loss when straw is covered. The larger the slope and intercept of the fitted straight line, It shows that the greater the ability of surface runoff to cause soil loss under straw mulching.
The title of Figure 5 in the manuscript is "division carrying capacity of soil". As you mentioned, this title is not appropriate. After our discussion, we decided to modify the titles of figures 5 and 6. The title of Figure 5 has been changed to " Relationship between surface runoff rate and surface soil loss rate in lime soil young citrus orchard under different straw coverage." The title of Figure 6 has been changed to " Relationship between surface runoff rate and surface soil loss rate in red soil young citrus orchard under different straw coverage."
Point 13: “surface runoff intensity” – usually, the term “surface runoff rate” is used.
“surface erosion rate” – usually, the term “soil loss rate” is used.
Response 13: Thank you for your suggestions on the revision of the manuscript. With regard to the improper use of some words in your manuscript, we have revised them all, and checked the whole article to keep the whole academic language consistent. Thank you again for your suggestions.

Round 2
Reviewer 1 Report
The authors succeded in answering to my concerns, so the article could be accepted,
Author Response
Point 1: The authors succeded in answering to my concerns, so the article could be accepted.
Response 1: Thank you for your recognition of the revised manuscript. In the revision of the manuscript, we have added some missing content and revised some academic words. In the subsequent revision, we sought MDPI professional English editing service to further improve the English expression ability of the article.

Reviewer 4 Report
Some parts of manuscript were improved, but still the manuscript contains a lot of grammar and stylistic errors and terminological inconsequence. In my opinion, further correction is necessary. Please, take some time and slowly read the text, analyze it, find and correct errors. I came (with an abrupt revision) to page 8, and then stopped. I did not mention about it previously, but there is no description of rainfall simulator. At least, a few sentences should be given by authors about type of simulator, drop size and height of falling drops.
In my opinion, the presented version of manuscript cannot be acceptable for publication and still requires a major revision and improvement.

Author Response
Point 1: Title is formulated in an improper way. What it means? I suggest: Effects of different rates of straw mulch on runoff and sediment yield in lime and red soils under young citrus plants in studies with a rainfall simulators”. It is one of some possibilities. Words: karst area, southwest China could be used as keywords.
Response 1: Thank you for your suggestions for revisions to the title and keywords of the manuscript. We fully accept your suggestion to revise the title. The new title emphasizes the "rain simulator", which is one of our main experimental setups. So we decided to revise the title according to your suggestion. The revised title is "Effects of Different Rates of Straw Mulch on Runoff and Sediment Yield of Young Citrus Orchards in Lime Soil and Red Soil in Studies with a Rainfall Simulator".
In the previous title, "karst area" and "Southwest China" appeared. In order to avoid repetition of keywords, so we didn't use these two keywords. After changing the new title, this contradiction no longer exists, so we replaced "ecological fragile zone" in the keyword with "karst area; southwest China".
Point 2: I still do not understand the meaning of term “soil stripping”. Is the term could mean soil detachment by raindrops or runoff?
Response 2: Thank you for asking questions about the vocabulary in the manuscript. "Soil stripping" refers to the process by which raindrops or runoff remove soil detachment from the surface. It mainly depends on the kinetic energy of raindrops and runoff. For example, the greater the runoff velocity, the greater the possibility of "soil stripping". This is a phenomenon associated with the occurrence of soil erosion.
Point 3: The keyword “ecological fragile zone” is so wide that it could be referred to everything.
Response 3: Thank you for your suggestion to revise the keywords in the manuscript. As you mentioned, "ecological fragile zone" is too broad, we have replaced this keyword with "karst area; southwest China ".
Point 4: “high economic value such as economic forest or citrus” – Please, omit the second word
“economic”, just forest or citrus; economic importance was mentioned before.
Response 4: Thank you for your suggestion to revise the language expression in the manuscript. We fully accept your suggestion for revision and have revised this sentence in the manuscript. It has now been changed to "high economic value such as forest or citrus". In the subsequent revision, we sought MDPI professional English editing service to further improve the English expression ability of the article.
Point 5: “Straw mulching has been proven to be a typical conservation tillage measure.” By definition, tillage is classified as a conservation one, if at least 30% of soil area is covered by plant residue or mulch . So, the sentence should be “Straw mulching is a one of typical measures in conservation tillage”
Response 5: Thank you for your suggested revisions to the manuscript. We have rechecked the definition of "conservation tillage" and have made corresponding changes in the manuscript based on your suggestion. In subsequent revisions, we will continue to grasp the expression of details.
Point 6: “a straw mulch of 5.0 t/hm²” – the proper unit is t/ha not “t/hm²”
Response 6: Thank you for suggesting revisions to the units of the manuscript. We fully accept your modification suggestion. After inquiring about the correct unit, we have checked and modified the units of the full text.
Point 7: “The changes of runoff and sediment yeilds largely depend on the original soil conditions and topographic characteristics”. Please, correct: yields, not “original” but antecedent (conditions before the rainfall)
“The study was performed in order to verify two main hypotheses: (1) straw mulching could reduce the surface runoff and soil erosion, and increase the underground runoff and subsurface flow, and (2) the effect of straw mulching on reducing surface erosion in lime soil and red soil was different.” Remark: “and increase the underground runoff and subsurface flow” could be omitted (it is not necessary, and there is still some problem with terminology; see division of flow in hydrology: surface flow, subsurface flow, groundwater flow or decide on displacement of “flow” by “runoff” and apply the same terms everywhere). Please, use the same tense in the second part – not: “was different” but could be different.
Response 7: Thank you for your suggested revisions to the manuscript. We used "original" in your proposed sentence, and after reconsideration we have replaced the wording in the manuscript with "antecedent". Indeed it is expressed in the ref that the prerequisites for runoff and sediment volume are soil characteristics and topographic conditions prior to rainfall.
We mentioned a point in the scientific hypothesis that "straw mulching increases soil interflow and underground runoff". In the karst area of southwest China, especially the lime soil area, the soil layer is shallow and the soil infiltration rate is high. Under some smaller rainfall intensities, the main forms of runoff are subsurface flow and groundwater flow. Excessive underground runoff can still cause underground erosion and even contaminate groundwater. Therefore, the amount of soil interflow and underground runoff is also a necessary indicator. However, in the conclusion part, we mainly verified "straw mulching could reduce the surface runoff and soil erosion", so we accept your modification suggestion and delete "and increase the underground runoff and subsurface flow".
We have revised the tense question in the second part of the hypothesis as you suggested.
Regarding your mention of terminology, we have unified the terminology everywhere in the manuscript as you suggested, and the unified terminology is "surface runoff, subsurface flow, groundwater flow".
Point 8: “2.2.1. Soil tank”. In my opinion, the “2.2.1. Soil trough” could be better. The term “trough” is used for tank or container that was adjusted to studies with rainfall simulator.
Response 8: Thank you for your suggestion to revise the English words in the manuscript. We accept your suggestion for revision, the meaning of "trough" is indeed more suitable for this article, so we replace "soil tank" with "soil trough" throughout the text.
Point 9: “The lower end of the soil trough was a V-shaped surface collecting trough, which and was
connected with the collecting bucket through a hose to collect runoff and sediment generated on
the surface”
Response 9: Thank you for your suggested revisions to the manuscript. We have removed the redundant part of this sentence and reorganized the language as you suggested.
Point 10: “Before filling into the soil tank, in order to prevent soil leakage from the six round holes at the bottom of the soil tank, cover the six round holes at the bottom of the soil tank with square water soluble fiber with side length of 10cm, and then fill the soil into the soil trank.”
Remark: sentence is too long. Please, divide the sentence in two parts and correct style.
I suggest to use “trough” instead of “tank”
Response 10: Thank you for your suggested revisions to the manuscript. As you mentioned, the sentence was too long, so we reorganized the language, shortened the sentence, and replaced "soil tank" with "soil trough" in the text.
The revised sentence reads: "To prevent soil leakage from the bottom of the soil trough during the filling process, the six circular holes at the bottom of the soil trough were covered with water-soluble fibers, and then filled with soil".
Point 11: “The samples (surface runoff, sub-surface flow, underground runoff) were received in plastic buckets at the same time, and the total weight was obtained by weighing immediately”
Remark: still there is inconsequence with terms. Please, compare the text and table 2. I would prefer groundwater flow than “groundwater runoff”. In some other part of the text the authors used “The samples (surface runoff, sub-surface flow, underground runoff)”. However, still there is an inconsequence between title of the table (runoff coefficients) and names of columns (surface runoff, subsurface flow, groundwater runoff).
Response 11: Thank you for your suggestion to revise the wording of the manuscript. After we searched for relevant information, we finally determined the three academic terms " surface runoff, subsurface flow, groundwater flow" and kept them unified throughout the text. Thanks again for your suggestions on the manuscript.
Point 12: “Under moderate rain intensity (60 mm/h), the reducing rates of 20%, 50% and 80% straw overage in lime soil were 49.2%, 41.6% and 51.5%, respectively compared with no straw coverage, and those in red soil young citrus orchard were -3.4%, 3.7% and 10.3% respectively.”
Remark: the structure of sentence unable to understand what the authors wanted to communicate.
Please, reconsider, the structure of sentence.
Response 12: Thank you for your suggested revisions to the manuscript. This sentence is indeed not clear enough, so we have reorganized the language and made changes where it was not clear.
The revised language is: "At 60mm/h rainfall intensity, the surface runoff reduction rates of lime soil young citrus orchards under different straw coverage rates were 49.2%, 41.6%, and 51.5%, and the surface runoff reduction rates of red soil young citrus orchards were -3.4%, 3.7%, 10.3%”.
Point 13: There is no description of rainfall simulator. At least, a few sentences should be given by authors about type of simulator, drop size and height of falling drops.
Response 13: Thank you for your suggestions for revisions to the content of the manuscript. We are indeed missing a description of the rain simulator in the text, which is the main device used in our research. We have now included a description of the rainfall simulator in the manuscript. Thank you again for your suggested revisions.

This manuscript is a resubmission of an earlier submission. The following is a list of the peer review reports and author responses from that submission.
Round 1
Reviewer 1 Report
I had the chance to review this article on its submission to Sustainability. Compared with that submission, this version is improved, but not enough.
The subject of the article is interesting, and it is linked to the objectives of the journal, however, there are many issues that have to be reconsidered.
The literature review is quite well discussed, and the research hypotheses are not clearly presented.
The structure of the article is not presented, so it is difficult to be understood and follow.
Again, a large amount of data are presented without mentioning the references (see 2.1 for instance, e.g. )
Please add as a limitation to your approach / or as future research needs more research on measures to be taken.
I recommend that on Conclusions, to be presented the clear answers to the formulated research hypothesis.
Author Response
Response to Reviewer 1 Comments
Point 1: The literature review is quite well discussed, and the research hypotheses are not clearly presented.
Response 1: Thank you for your affirmation of the literature review in the manuscript. In the 'Introduction' of the manuscript, we further modified the research hypothesis, mainly from the effect of straw mulching measures on reducing flow and sediment in red soil and lime soil citrus orchards. The specific modifications are as follows:
We carried out this study in order to verify two main hypotheses: (1) straw mulching could re-duce the surface runoff and soil erosion, and increase the underground runoff and soil runoff, and (2) the effects of straw mulching on reducing surface erosion in lime soil and red soil was different.
(The modified part has been highlighted in the manuscript.)
Point 2: The structure of the article is not presented, so it is difficult to be understood and follow.
Response 2: Thank the reviewers for their comments on the structure of the article. After my rethinking, I think the main reason why the structure of the article is not presented is that the structure of the 'Introduction' part is chaotic. Therefore, I revised the 'Introduction' part of the manuscript again. You can view the 'introduction' part I edited again in the revised manuscript. Now we have revised the problems of language expression throughout the whole article, and changed some language expressions. In the follow-up writing, we will pay more attention to the problem of language expression, thank you again for your comments on the manuscript.
Point 3: Again, a large amount of data are presented without mentioning the references (see 2.1 for instance, e.g. )
Response 3: Thank you for paying for the lack of references in the manuscript. The references of longitude, latitude, temperature and other survey data of the study area in part 2.1 of the manuscript are:
Liu H.Q. Influence of bedrock ups and downs and karst pipeline morphology on the process of soil erosion in karst area. Master's degree, Huazhong Agricultural University, Wuhan China. 2020. (In Chinese)
Point 4: Please add as a limitation to your approach / or as future research needs more research on measures to be taken.
Response 4: Thank you for your suggestions for revisions to the manuscript, we have made corresponding revisions based on your suggestions. The last paragraph of the 'discussion' section of the manuscript discusses the limitations of this experimental method/ or as future research needs more research on measures to be taken. The details are as follows: Finally, the results of our study are obtained through the artificial rainfall test simulating rainstorm. In order to better study the runoff and sediment production characteristics of young orange orchards covered with straw in karst areas of Southwest China, it is necessary to further observe and study the runoff and sediment yield characteristics of young orange orchards through natural rainfall, and further explore its soil nutrient loss in the process of rainfall.
Point 5: I recommend that on Conclusions, to be presented the clear answers to the formulated research hypothesis.
Response 5: Thank you for your suggestion, we have made the appropriate changes based on your suggestion. We presented the research hypothesis in the "Introduction" section of the manuscript, validated it with experimental data, and now have a clear answer to the research hypothesis in the "Conclusion" section of the manuscript.
The main conclusions are as follows: Straw mulching measures can significantly reduce surface erosion in both lime soil and red soil from young citrus or-chards. The result proved with the first hypothese. In lime soil, the surface sediment reduction rate after straw mulching can reach 35.1%-94.2%. In the red soil, the surface sediment reduction rate under heavy rain (120 mm/h) with 50% straw coverage can achieve a significant surface erosion reduction effect of 24.7% - 97.9%. The effects of straw mulching on reducing surface flow in lime soil was significantly higher than that in red soil, which vertified the second hypothese. The surface flow reduction rate of the lime soil after straw mulching can reach 2.4%-51.5%, but the red soil was only -5.4%-10.3%. The main reason for reducing surface erosion of lime soil by straw mulching is limited runoff transportation, while for red soil is decrease of soil stripping.

Reviewer 2 Report
The authors quantified the lumped effect of rice straw cover on runoff and soil loss in a lab experiment. This already implies that the title of the manuscript is misleading because neither young citrus orchards played a role nor was the situation on the Chinese Loess Plateau analyzed.
I cannot support this manuscript because (1) a scientific question is missing. (2) The methods remain unclear and do not allow repeating the experiment, assessing their relevance for the research question and interpreting the results. (3) The language has large deficits. I will illustrate/explain these three reasons below.
- There is hardly any other influence on soil erosion by water that has been analyzed in more experiments and is understood in greater detail than that of a straw cover. In particular, we understand its interaction with other soil properties like soil roughness and we understand its direct effects preventing raindrops from hitting the soil surface or decelerating runoff and its indirect effects via the increase in soil moisture, microbial activity or earthworms. Importantly for the manuscript, we also understand since more than 50 years the interaction with vegetation cover (which surprisingly was not measured in this case although the description lets me assume that some plant cover was present). In contrast to this deep knowledge, the authors claim that "the benefits... are still unclear" without explaining what remains unclear and without measuring all important properties like soil roughness, soil moisture, vegetation height and vegetation cover that would be required to find out whether the measured values agree with present knowledge or not. My impression was that the benefits were simply unclear to the authors. This is, however, not a scientific question and it cannot be overcome by an experiment but requires studying the relevant literature by the authors of this manuscript.
- The methods are unclear and incomplete and thus it cannot be assessed whether they are appropriate. It is unclear what a red soil or a lime soil is. It is unclear which properties the rain had. It is unclear what 'replicates' mean in this case. It is unclear, which statistical methods were applied. The many soil properties and vegetation properties, which are required to understand the direct and indirect effects of the mulch cover on soil loss and runoff, have not been reported.
- The English is insufficient. In particular, the authors do not know the basic technical terms (which is not surprising given their lack of knowledge of the international literature). The style is that of a manual for a subordinate technician using the imperative but it is not the style of a publication. It is just impossible to understand what the authors mean. One example:
"Before loading soil, lay a sheet on the reserved hole of the opening 10 x 10 cm water-soluble cotton (soluble in water), and then fill."
What is 'loading soil'? What is a 'reserved hole'? Which opening? Several openings were described earlier. No opening was 10 × 10 cm. Cotton is never soluble in water. Fill what with what?
Author Response
Response to Reviewer 2 Comments
Point 1: The authors quantified the lumped effect of rice straw cover on runoff and soil loss in a lab experiment. This already implies that the title of the manuscript is misleading because neither young citrus orchards played a role nor was the situation on the Chinese Loess Plateau analyzed.
Response 1: Thank you for your comments, we have revised the title of the article according to your suggestions. The main question of our study is the effect of straw mulching measures on reducing runoff and sediment in young citrus orchards in red soil and lime soil in Southwest China. A literature review of straw mulching measures was also carried out. After reconsidering the main significance of the research, I have revised the title of the manuscript to "Reduction of Straw Mulching on Runoff and Sediment of lime soil and red soil in Young Citrus Orchards in Karst Area of Southwest China"
Point 2: a scientific question is missing.
Response 2: Thank you for your suggestions for revisions to my manuscript. As you mentioned, the scientific issues described in the manuscript are really not clear enough. We have made corresponding modifications in the manuscript according to the modification suggestions you put forward. Soil and water loss is easy to occur in the early stage of citrus planting. This study is to research the effect of straw mulching on reducing surface runoff and sand production in the early stage of citrus planting. The scientific question has now been described in the ‘Introduction’ section of the manuscript, with the following specific revisions: ‘Citrus orchard is common in karst areas of Southwest China. However, at the beginning of citrus planting, the surface of the orchard is bare and the serious soil erosion has become one of the main obstacles to agriculture development. Tu A.G conducted long-term observations of soil erosion in citrus orchards, and the results showed that soil erosion intensity in citrus orchards can reach extremely strong levels during the young tree stage [3]. The runoff coefficient was above 0.3 and the soil erosion was serious at the beginning of citrus planting in karst area of Spain [4]’.
Point 3: The methods remain unclear and do not allow repeating the experiment, assessing their relevance for the research question and interpreting the results.
Response 3: Thanks to the comments made by the reviewers, we have re described the test methods, including the collection methods of surface runoff, soil flow, underground runoff and surface sediment. Our research is based on the artificial simulated rainfall test. If only one experiment is conducted under each control variable under the simulated rainstorm intensity, the uncertainty of the experimental results is too large to ensure the scientificity and universality of the experimental data. Therefore, we have conducted three repeated experiments on each control variable, Finally, the data used in our manuscript is the average of the three repeated test data, which makes the test results more objective and improves the credibility of the test results.
Point 4: The language has large deficits.
Response 4: Thank you for your suggestion on the inadequacy of the language expression of the manuscript, some of which were indeed quite deficient in the initial submission of the manuscript. Now we have revised the problems of language expression throughout the whole article, and changed some language expressions. In the follow-up writing, we will pay more attention to the problem of language expression, thank you again for your comments on the manuscript.
Point 5: There is hardly any other influence on soil erosion by water that has been analyzed in more experiments and is understood in greater detail than that of a straw cover. In particular, we understand its interaction with other soil properties like soil roughness and we understand its direct effects preventing raindrops from hitting the soil surface or decelerating runoff and its indirect effects via the increase in soil moisture, microbial activity or earthworms. Importantly for the manuscript, we also understand since more than 50 years the interaction with vegetation cover (which surprisingly was not measured in this case although the description lets me assume that some plant cover was present). In contrast to this deep knowledge, the authors claim that "the benefits... are still unclear" without explaining what remains unclear and without measuring all important properties like soil roughness, soil moisture, vegetation height and vegetation cover that would be required to find out whether the measured values agree with present knowledge or not. My impression was that the benefits were simply unclear to the authors. This is, however, not a scientific question and it cannot be overcome by an experiment but requires studying the relevant literature by the authors of this manuscript.
Response 5: Thank you for your comments on the manuscript, we have carefully revised the manuscript based on your comments. The main purpose of this study was to investigate the effect of reducing runoff and sediment after straw mulching measures in young citrus orchards with two soil types in Southwest China. In the "Introduction" of the manuscript, we made a literature review on the effect of straw mulching on abortion and sediment production, and clarified that straw mulching measures are very effective measures for soil and water conservation. 'the benefits... are still unclear' means that the effect of straw mulching measures on stream reduction and sediment reduction in young citrus orchards in southwest China is unclear.
The 'soil roughness and soil moisture' mentioned in your revision suggestion are very important indicators, but in the current experiment, there is no focus on measurement, but runoff and erosion are used as important indicators to characterize 'the benefits' . In the next stage of the experiment, we will further study 'soil roughness and soil moisture' based on your suggestion.
Point 6: The methods are unclear and incomplete and thus it cannot be assessed whether they are appropriate. It is unclear what a red soil or a lime soil is. It is unclear which properties the rain had. It is unclear what 'replicates' mean in this case. It is unclear, which statistical methods were applied. The many soil properties and vegetation properties, which are required to understand the direct and indirect effects of the mulch cover on soil loss and runoff, have not been reported.
Response 6: I am very grateful to the reviewers for their comments on my manuscript. I lack the explanation of 'Red Soil' and 'lime soil' in the manuscript, which will be supplemented now.
Red soil usually has deep red soil layer, with obvious development of reticulate layer. Clay minerals are mainly kaolinite, acidic and low base saturation. The red soil is not easy to be damaged by rain erosion, so the red soil develops well under the leaching of rain.Lime soil is the soil developed by limestone parent material. The texture of lime soil is generally sticky, and there are more or less lime foam reactions on the section. It often has red, yellow, brown, black and other colors. China's lime soil is mainly distributed in the southern subtropical region.
During our experiment, the experiment under each rainfall intensity and straw coverage will be repeated for 3 times, and the final experimental data will take the average value of the experiment repeated for 3 times. In this study, IBM spssstatistics25 and origin2017 are used to statistically analyze the simulated rainfall runoff data and sediment data collected in the experiment, including significance analysis and data fitting.
Point 7: "Before loading soil, lay a sheet on the reserved hole of the opening 10 x 10 cm water-soluble cotton (soluble in water), and then fill."
What is 'loading soil'? What is a 'reserved hole'? Which opening? Several openings were described earlier. No opening was 10 × 10 cm. Cotton is never soluble in water. Fill what with what?
Response 7: We appreciate your pointing out the misuse of certain words in the manuscript. Because of my English expression mistakes, I used wrong words in the manuscript, making the meaning of some words misleading. After re-checking the terminology, the erroneous words have now been replaced in the manuscript.'loading soil' refers to the process of filling the soil used for planting orange trees with ‘the adjustable slope steel soil trough’ during the test.
'reserved hole 'refers to the 6 holes with a diameter of 5cm at the bottom of' the adjustable slope steel soil trough '. The function of' reserved hole 'is to collect underground runoff.
Since there are six holes evenly arranged at the bottom of the soil tank, in order to ensure that the test soil does not leak from the six holes at the bottom of the soil tank during the filling of the soil tank, we cut the 'water-soluble fiber' (the main material of this fiber is PVA fiber, which will dissolve immediately after rain) into a square with a side length of 10cm, Cover the six holes at the bottom of the soil tank. When all the soil is filled into the soil tank, when the soil moisture content reaches a certain level, 'water-soluble fiber' will be dissolved by water and will not affect the collection of underground runoff. Due to the wrong expression in English, I express' water-soluble fiber 'as' Cotton'. The previous incorrect expression of English vocabulary has been corrected in the manuscript.

Round 2
Reviewer 1 Report
Dear authors, thank you for improving the article, now on it looks much better compared with its first submission.
For better visibility on databases, the authors are asked not to repeat among keywords the words/concepts included in the title of the article.
Reviewer 2 Report
The authors have expanded the review on current understanding of the effects of straw mulch but they still failed to provide a new scientific question. The sole justification for their research is that few studies considered straw mulching in soil and water conservation in citrus orchards of karst areas. This may be highly relevant to justify local demonstration experiments for stakeholders but this cannot be a justification for a scientific experiment, which requires a hypothesis based on current knowledge that could be tested to provide new insights.
Also the description of the experiment is still impossible to understand and full of linguistic mistakes (Example: “Beforefilling soil into the soil tank,a sheet of the opening 10×10 cm water-soluble fiber (soluble in water) was layed on the reserved hole, and then fill the soil.”) Hence, it is impossible to assess whether the experimental procedures were adequate and reliable.
I still cannot support this manuscript.